# Precarious Work as Risk Factor for 5-Year Increase in Depressive Symptoms

**DOI:** 10.3390/ijerph19063175

**Published:** 2022-03-08

**Authors:** Yucel Demiral, Tobias Ihle, Uwe Rose, Paul Maurice Conway, Hermann Burr

**Affiliations:** 1Department of Public Health, Medical Faculty, Dokuz Eylül University, Izmir 35340, Turkey; yucel.demiral@deu.edu.tr; 2Federal Institute for Occupational Safety and Health (BAuA), 10317 Berlin, Germany; tobias.ihle-potsdam@web.de (T.I.); rose.uwe@baua.bund.de (U.R.); 3Department of Psychology, University of Copenhagen, 1165 Copenhagen, Denmark; paul.conway@psy.ku.dk

**Keywords:** precarious work, non-standard work, mental health, prospective analyses

## Abstract

Objectives: The aim was to investigate the longitudinal relationship between precarious work and depressive symptoms in a representative cohort of employees in Germany. Methods: In the German Study on Mental Health at Work (S-MGA) (*n* = 2009), depressive symptoms were assessed by the Patient Health Questionnaire (PHQ−9). Precarious work was measured through baseline (2012) self-reported job insecurity, marginal part-time, fixed-term contract, hourly wage and—during follow-ups 2012–2017—unemployment. Among employees without depressive symptoms at baseline (2012), we ran logistic regression analyses stratified by gender with depressive symptoms at follow-up in 2017 as the dependent variable, adjusting for baseline (2012) age, gender, socioeconomic position and partner status. Results: Among men, job insecurity (OR: 2.47; 95% 95% CI: 1.37–4.48) and low wage (3.79; 1.64–8.72) at baseline were significantly associated with depressive symptoms at follow-up. Among women, indicators of precarious work were not associated with depressive symptoms at follow-up. Among men, a cumulative exposure index of precarious work was significantly associated with the development of depressive symptoms (one indicator: 1.84; 0.94–3.60, ≥two indicators: 7.65; 3.30–17.73). This index was not associated with depressive symptoms among women. The population attributable fraction of precarious work due to depressive symptoms among men was approximately 30%. Conclusions: Among employees in Germany, precarious work seems to be a risk factor for the subsequent development of depressive symptoms among men, but not among women. Research on precarious employment in different countries is needed.

## 1. Introduction

The employment policies associated with the Keynesian economy, which became the norm in the “Golden Age” that followed World War II, ended with the global economic crisis in the 1970s. As a response to this, flexible employment schemes have been introduced along with restrictions in social protection programs [1]. Since then, there has been a wide range of flexible work practices under different names, including piecework, casual work, contingent work and so forth [1]. In Germany, deregulations have led to a rise of employment in fixed-term contracts, agency work and marginal part-time work [2,3,4,5]. In a European comparison, employees in Germany showed an average level of job insecurity but one of the highest fractions of low wage earners.

Several scholars have brought forward the notion of precarious work in order to grasp the increased flexibilization of work in light of its possible consequences for health [6,7]. Most definitions of precarious work have the concept of insecurity in employment in common [8,9,10,11,12]. This concept includes two aspects. The first indicates the presence of insecure contracts (e.g., fixed-term, temporary agency or lack of contract) [8,9,10,13]. The second relates to the experience of job insecurity [11], reflecting the subjective anticipation of risk of unemployment, due to, for instance, low formal employment security or economic difficulties of the employer. Therefore, some scholars also include insecurity about income level as part of the definition of precarious work [1,14]. Here, precarious work arises due to economic hardships in maintaining a household [9,10,11,12,13]. It has been suggested that definitions should also include the multifaceted issue of insufficient legal rights and their enforcement in practice related to employment and wage as this aspect might contribute to the experience of insecurity [8,11,12,13]. Some scholars define precarious work as being subjected to a poor work environment [10,11,12]; others suggest that a poor work environment should rather be regarded as a result of precarious employment [7]. In the present paper, we only focus on precarious work defined through extrinsic employment conditions such as the employment contract and pay, leaving out intrinsic aspects related to working conditions such as demands or resources at work.

From a theoretical perspective, stress is assumed to play a key role in the relationship between precarious employment and health [15], especially depressive symptoms [1]. The sustained stress that individuals might endure in connection with precarious work is due to experiences of insecurity regarding their employment and employment conditions. These experiences may derive from a perceived threat of losing one’s job and/or from the fear of being unable to sustain a living based on the uncertain continuation of one’s employment [14,16,17]. Such an impact can be explained by the central role that work plays in fulfilling fundamental human needs, including the needs for survival, autonomy, belongingness and competence, which are severely threatened when individuals fear losing their job [18]. 

Based on literature searches performed on Medline and EbscoHost in the period 2020–2021, as well as forward and backward citation searches on Web of Science and Google Scholar, we found 22 longitudinal studies and one meta-analysis covering 11 of these studies, investigating the association between precarious work and depressive symptoms [19,20,21,22,23,24,25,26,27,28,29,30,31,32,33,34,35,36,37,38,39,40] (Table 1 and Table 2). We only focused on longitudinal studies as these are more effective in establishing causality than cross-sectional studies [41]. Of these, 11 were of Scandinavian origin, six were based on data from continental Europe and five on data from other continents (North America, Asia, Australia). Of the 22 studies we identified, six investigated combined measures of precarious work [19,20,21,22,23,24]; 16 studies focused on specific dimensions, mainly job insecurity, which was examined in 14 studies [19,21,22,26,27,28,29,30,31,35,36,37,38,39,40]; however, only three studies examined fixed-term contracts [32,33,34]; and only two unemployment [22,25] (as one study considered more than one factor, the studies do not add up to 22 [22]). To our knowledge, no prospective studies exist investigating the effects of marginal part-time (German: ‘Mini- or Midijob’) or low wage. Importantly, ten studies (five from Scandinavia and five from other countries) have previously assessed if the effects of precarious employment differed between men and women.

In the present study, we focused on five aspects of precarious work as risk factors for depressive symptoms among workers in Germany: (a) job insecurity, (b) working in a fixed-term contract, (c) marginal part-time, (d) low wage and (e) unemployment. We hypothesized that specific as well as concurrent experiences of precarious work elevate the risk of depressive symptoms. We also examined possible gender-related differences in these associations.

## 2. Materials and Methods

### 2.1. Sample

The present study is based on data from the German Study on Mental Health at Work (S-MGA): a nation-wide employee cohort study with a baseline in 2012 and a follow-up in 2017 [43]. At baseline, the population consisted of all employees in Germany aged 31–60 years included in a social security scheme set up by the government (i.e., without civil servants, self-employed individuals and freelancers) on the reference date of the 31 December 2010. Of the 13,590 employees who were randomly selected from 206 municipalities in Germany (stratified by region and population size), 4511 people completed the baseline personal interview that was conducted 13 months (range 11–17 months) after the sampling procedure was completed (baseline response rate: 33%) (Figure 1). During these 13 months, 308 people ceased to be employed, leaving 4203 employed participants at baseline; of these, 2485 also participated in the follow-up interview (follow-up participation among baseline participating employees was 59%, while the cohort participation was 20%; see Table 3). Cohort participation was independent of gender, but was dependent on age and socioeconomic position (SEP): participants 30–39 years of age and in a lower socioeconomic position had lower participation (Table 3). From these, 313 people were excluded, predominantly due to missing information on depressive symptoms. For the main analysis, 163 participants with depressive symptoms at baseline were excluded (see analysis section below). Thus, the final sample on which the present study is based amounts to 2009 employees. 

### 2.2. Measures

All information, with only the exception of depressive symptoms, was obtained through interviews conducted in the respondents’ homes [43]. Information on depressive symptoms was obtained through a paper questionnaire filled in by the participants in the absence of the interviewer [43].

#### 2.2.1. Outcome Measure: Depressive Symptoms

The Patient Health Questionnaire (PHQ–9) was used to measure depressive symptoms at both baseline and follow-up [44]. The scale consists of the following nine items, measuring the intensity of symptoms as experienced by the respondent within the past two weeks: ‘Little interest or pleasure in doing things’; ‘Feeling down, depressed or hopeless’; ‘Difficulty falling asleep or sleeping or increased sleep’; ‘Tiredness or lack of energy’; ‘Decreased appetite or excessive need to eat’; ‘Bad opinion of yourself’; ‘Difficulty concentrating on something’; ‘Slowed speech/movement or restlessness (“fidgety”)’; and ‘Thoughts that you would rather be dead or want to inflict pain on yourself’. The response categories are: ‘Not at all’ (0), ‘Several days’ (1), ‘More than half the days’ (2), and ‘Nearly every day’ (3). The scale for depressive symptoms was calculated as the sum of the nine items, resulting in a score ranging from 0 to 27 [44,45]. Cronbach’s α was 0.82, with inter-item correlations ranging between 0.20 and 0.49. For the main analysis, the scale was dichotomized using ≥10 as the cut-off point, which represents a clinically validated cut-off for depression cases [46]. In the sensitivity analysis, the scale was treated as a continuous measure.

#### 2.2.2. Precarious Work

Precarious work was measured through five indicators, of which four were measured at baseline (job insecurity, marginal part-time, fixed-term contract, low wage), and one at follow-up (events of unemployment during follow-up). 

Self-reported job insecurity in 2012 was measured as a scale calculated as the mean of the following two items from the job insecurity scale [47,48]: ‘Are you worried about becoming unemployed?’ and ‘Are you worried about it being difficult for you to find another job if you became unemployed?’. These included the response options: ‘To a very large extent’ (4), ‘To a large extent’ (3), ‘Somewhat’ (2), ‘To a small extent’ (1) and ‘To a very small extent’ (0). Cronbach’s α was 0.52, with an inter-item correlation of 0.35. The job insecurity scale was dichotomized, with a score of 2.5 or higher indicating job insecurity.

Marginal part-time (German term: ‘mini-/midi-job’ [5,49]) in 2012 was assessed as a dichotomous variable based on the questions: ‘What is your main activity?’ and ‘Are you currently …?’, with the response option ‘Marginally, occasionally or irregularly, also in mini-jobs up to 400 euros or midi jobs up to 800 euros a month’ considered as an indicator of marginal part-time. All the other options, e.g., ‘Working full time at least 35 h per week’ and ‘Unemployed’, were not regarded as marginal part-time. Marginal part-time covers a type of contract introduced in the German employment legislation called ‘minor employment’, characterized by part-time employment with a comparatively low income and limited obligations for the employer regarding social security contributions [5,49].

Fixed-term contract in 2012 was measured as a dichotomous variable based on the question: ‘Is your current employment contract …?’, with the two response options being ‘fixed term’ and ‘open ended’.

Low wage in 2012 was measured as a dichotomous variable; participants with low wages were considered as those obtaining below 60% of the median personal hourly net wage in the present cohort (i.e., below €6.4/$7.3). We calculated hourly net wage based on two questions: one regarding current monthly net income (calculated as a scale) and one regarding working hours. Current monthly net income was divided by working hours, and the resulting variable was used as the basis for the dichotomization. Information on wage was calculated as a scale based on the question: ‘Please report how much of your current monthly net income comes from your professional activity’, with the following response options: ‘Up to €400 ($450)’, ‘Over €400 to €800 (>$455–910)’, ‘Over €800 to under €1000 (>$910–<1135)’, ‘€1000 to under €1500 (1135–<$1705)’, ‘€1500 to less than €2000 (1705–<$2270)’, ‘€2000 to less than €2500 (2270 < $2840)’, ‘€2500 to less than €3000 (2840 < $3405)’, ‘€3000 to under €4000 (3405 < $4540)’, ‘€4000 to less than €5000 (4540–<$5675)’ and ‘€5000 and more (≥$5675)’. Each of these response options was coded based on the midpoint of the respective range of income. Information on working hours was collected with the following two questions: ‘How many hours a week do you normally work, including regular overtime, overtime, etc.?’ and–for people with a second job(s)–‘How many hours do you usually work there each week?’. Weekly working hours were recalculated into monthly working hours.

Unemployment between 2012 and 2017 was coded as a dichotomous variable based on the following questions at follow-up, inquiring about episodes of unemployment that occurred since baseline: ‘What was your main activity then?’. With one of the response options being ‘Unemployed’, all the other options, e.g., ‘Working full time at least 35 h per week’ and ‘Marginally, occasionally or irregularly, also in mini-jobs up to 400 euros or midi jobs up to 800 euros a month’, were regarded as not indicating unemployment [50]. We constructed a dichotomous variable reflecting experiences of at least one spell of ≥1 month with unemployment between baseline in 2012 and follow-up in 2017.

A summary index including all indicators of precarious work was constructed (precarious work index); only the indicators being significantly associated with depressive symptoms at follow-up in at least one gender were included in the index [40]. As the number of employees experiencing more than two indicators was small, the highest category of the index was ‘≥2 indicators’, with 6% (118 people) experiencing at least two indicators at the same time, 25% (498 people) only one indicator and 69% (1393 people) none of the indicators. 

#### 2.2.3. Covariates

Information about participants’ gender and age was obtained through the interview. Squared age was introduced in the analysis as the incidence of depression increased up to the start of the forties and then decreased slightly in the fifties [51,52]. 

The socioeconomic position (SEP) was assessed according to the four-level International Standard Classification of Education (ISCED), based on the International Standard Classification of Occupations 2008 (ISCO) [53]. Managers were placed in the same group as professionals in accordance with similar classifications [54]. The main groups of ISCO 2008 1 and 2 were placed in skill level category 4, the main group 3 in level 3, the main groups 4–8 in level 2 and the main group 9 in level 1.

Partner status was assessed by means of questions on cohabitation and partner status.

### 2.3. Analysis

In the main analyses, which include employees without depressive symptoms at baseline (PHQ9 < 10, see description of this variable above), the gender-stratified associations between dimensions of precarious work at baseline (job insecurity, fixed-term contract, marginal part-time, low wage and unemployment) and depressive symptoms at follow-up were examined by means of a multiple logistic regression, with estimates calculated as odds ratios (OR) and their relative 95% confidence intervals (95% CI). We estimated two models, each comprising two steps. In the first model, each dimension of precarious work was entered separately. In the first step, we adjusted for the baseline covariates age and partnership status. In the second step, we additionally adjusted for the socioeconomic position (SEP, entered as categorical variable). The second model followed the same procedure, except that all dimensions of precarious work were entered simultaneously in both steps.

We then examined the association with the precarious work index (see description above) at baseline and depressive symptoms at follow-up, following the same adjustment procedure described above. Based on this index, the population attributable fraction (PAF) of depressive symptoms due to precarious work was computed according to the method developed by Miettinen [55]. PAF can be understood as the fraction of depressive symptoms at follow-up attributable to precarious work. It can be illustrated graphically as the fraction of the area of bars (surplus cases) over the odds ratio 1 out of the total area of bars (total cases) [55].

In a first set of sensitivity analyses (Appendix A), we repeated the main analyses while treating depressive symptoms as a continuous, instead of dichotomous, variable in multiple linear regressions. In these analyses, the change in depressive symptoms was treated as the dependent variable, adjusting for baseline depressive symptoms, partnership status, age and–in a consecutively adjusted model–socioeconomic position. This was done to check if the estimates obtained in the main analyses were dependent of how the depressive symptoms’ variable was treated (dichotomized versus continuous). 

In a second set of sensitivity analyses (Appendix A), we repeated the main analyses while also including employees with depressive symptoms at baseline (PHQ–9 ≥ 10), adjusting for baseline depressive symptoms, partnership status, age and–in a consecutively adjusted model–socioeconomic position. This was done to check the impact on the results due to excluding participants with depressive symptoms at baseline who might have already developed these symptoms as a result of precarious work [56]. 

## 3. Results

Table 4 shows the distribution of the sociodemographic characteristics and the indicators of precarious work for the study sample of employees aged 31–60 years at baseline. Among the participants, 35% were exposed to some type of precarious work. Women reported significantly more marginal part-time and low wage than men. However, the prevalence of job insecurity and fixed-time work was not statistically different between genders. Job insecurity significantly increased with age. In general, all indicators of precarious work were strongly correlated with each other, with a prevalence two to three times higher in the exposed than in the unexposed group. Only marginal part-time and job insecurity were not significantly associated. In all, 5% (*n* = 54) developed depressive symptoms among male workers, and 9% (*n* = 89) among female workers (Table 5 and Table 6)

As shown in Table 5, among male employees without depressive symptoms at baseline, job insecurity (odds ratio (OR): 2.47; 95% confidence interval (95% CI): 1.37–4.48) and low wage (OR: 3.79; 95% CI: 1.64–8.72) at baseline were significantly associated with depressive symptoms at follow-up when entered separately in the regression model and after adjustment for age, partnership status and SEP. The associations also remained significant, despite being slightly attenuated, in the model wherein all indicators of precarious work were entered simultaneously. The associations were all non-significant among women (Table 6). Among employees without depressive symptoms at baseline, the experience of unemployment between baseline and follow-up was associated with depressive symptoms at follow-up among men but not among women (OR: 3.07; 95% CI: 1.28–7.37; Table 7). The associations were similar when including employees with depressive symptoms at baseline (Table A6, Table A7 and Table A8). 

When treating depressive symptoms as a continuous variable, job insecurity at baseline were associated with increased depressive symptoms among men, whereas low wage was no longer significant (Table A3 and Table A4). No other indicators were significantly associated with depressive symptoms among either men or women (Table A3, Table A4 and Table A5).

Table 8 shows the associations between the precarious work index at baseline and depressive symptoms at follow-up. The index included job insecurity, low wage and unemployment, as these were the indicators significantly associated with depressive symptoms in at least one gender. Among men without depressive symptoms at baseline, the precarious work index was associated with subsequent depressive symptoms (exposure to one indicator, OR: 1.84; 95% CI: 0.94–3.60; concurrent exposure to ≥2 indicators, OR: 7.65; 95% CI: 3.30–17.73). Among women, the associations were non-significant. The sensitivity analysis, including employees with depressive symptoms at baseline, showed similar results (Table A9). If one disregarded low wage in the precarious work index, the odds for both one and two indicators dropped to around 2 (table not shown). 

Based on the estimates of the precarious work index (Table 8), the PAF (population attributable fraction) of depressive symptoms due to precarious work was 32% among men. In the male sample, including those with depressive symptoms, the PAF was 27%. Figure 2 presents an illustration of this latter PAF. If we disregarded unemployment during follow-up in the index, the PAF was still around 30% among men when including all indicators of precarious work (data not shown). If one disregarded low wage from the index, the PAF dropped to 21% (data not shown).

## 4. Discussion

The present prospective study of employees in Germany aged 31–60 years suggests that precarious work is a risk factor for depressive symptoms, and that such a risk is higher among men than women. The odds of depressive symptoms were three times higher among men with unemployment and low wage, and two times higher among men with job insecurity. With regards to women, however, we did not observe any significant associations between precarious work and depressive symptoms. 

Among men exposed to two or more indicators of precarious work, the odds were five times higher. Our study indicates that the PAF (population attributable fraction) of depressive symptoms due to precarious work among employed men in Germany is around 30%.

It has been suggested that uncertainty about one’s ability to sustain a living is a key explanation for the role of precarious work in the development of depressive symptoms [57,58]. In the present study, job insecurity and unemployment predicted depressive symptoms among men; these could therefore be considered as the key sources of uncertainty among male employees. Multiple studies carried out in European countries generally support our finding that precarious work has a stronger effect on depressive symptoms among men than among women [20,25,26,27,28,29,31,42]. It should be noted that, in several studies, the effects for women were statistically significant but of a lower size than those observed among men. There might be a number of possible explanations for such a gender difference. One could be that in Germany–as in the case of other countries wherein studies about precarious work were performed–men contribute to the household’s income to a higher degree than women [59]. For this reason, precariousness can be expected to be a stronger stressor among men because of their typical role as breadwinners. We are only aware of one longitudinal study that has examined such a role [31]; in this study, the authors found that the risk of depressive symptoms due to job insecurity was elevated only among men who were contributing the most to the household, but not among other men (no clear results could be obtained among women due to the lack of power). Another explanation could be that men tend to identify more strongly with their work role than women do, whereas women identify more with family obligations [60]. We are not aware of more recent explanations based on work role. In addition, it might be that men and women have different preferences in relation to precarious work. For example, for some workers, fixed-term contracts offer an opportunity to collect various work experiences in order to gain skills and experience [61]. However, in Germany there is no gender difference regarding the voluntary choice of fixed-term contracts [62]. We are not aware of data on the voluntary choice of other types of precarious work among German employees.

### 4.1. Comparison with Previous Studies

In the following, we restrict the comparison of the results obtained in the present study with those obtained in other prospective studies only [41,63].

#### 4.1.1. Precarious Work as a Global Measure

Our finding that the concurrent exposure to indicators of precarious work was associated with an elevated risk of depressive symptoms among men corroborates the results obtained in a number of previous studies, including a study on the German SOEP cohort [19,20,21,22,23,24] (Table 1). In the present study, the OR related to the exposure to at least two indicators of precarious work was markedly higher that the risk observed in the studies included in the comparison. None of these studies considered low wage as indicator; when in the present study we excluded low wage in the calculation of the precarious work index, we found ORs of the same magnitude as in the other studies. One study on Swedish workers, which defined precarious work through three indicators (unemployment, temporary employment and perceived job insecurity), found that 18% of all cases of poor mental health could be attributed to precarious work. This fits with our estimate of 20% when not considering low wage as an indicator; our estimate rose to around 30% when low wage was also considered. Calculations of attributable fractions are advantageous as they can form a basis for prioritization of public health measures. Obviously, one should take into account to what extent prevention is feasible and if the associations observed are of a causal nature.

#### 4.1.2. Specific Indicators of Precarious Work

Regarding the effects of job insecurity on depressive symptoms, a meta-analysis of thirteen longitudinal studies–including the above-mentioned German SOEP study [22]—found an average elevated risk [40] (Table 2, first columns). In three studies not considered in this meta-analysis, the observed risks were of a similar size [22,28,31]. In some of these studies, the risk was higher among men than among women [26,27,28,29,31]. Overall, these findings are in line with the present study, wherein we found an OR of 2.47 (95% CI: 1.37–4.48) among men and of 1.41 (95% CI: 0.83–2.40) among women.

Regarding fixed-term contracts, three studies examined the effects on mental health; among these, two found an elevated risk while one did not observe any significant associations [32,33,34] (Table 2, middle columns). In the latter study, the outcome was sickness absence with depressive symptoms as the diagnosis, which might explain the non-significant association; it might be that diagnosed depressive symptoms do not reflect the underlying prevalence of disease. None of these studies were stratified by gender. The fact that in our study we did not detect a significant risk could be due to lack of power.

Regarding unemployment, we are aware of two studies that found an elevated risk for depressive symptoms, namely the above-mentioned German SOEP study and a Swedish one [22,25] (Table 2, last columns). In particular, in line with the present study, the latter found a higher risk among men than among women.

To our knowledge, no previous longitudinal studies have examined the effect of marginal part-time (German: “mini- or midi-job”) or low wage on mental health (see also Section 2.2.2, ‘Precarious work’, in the present paper).

#### 4.1.3. National Welfare State Context

National welfare state contexts may play a role in the association between precarious work and mental health. High levels of social protection might buffer the effects of precarious work on health [1,64]. The developed German welfare state might have played such a buffering role; although, since the turn of the millennium, it has restricted its level of social protection [2,3,4,5]. As the number of prospective studies is limited–especially outside Scandinavia–it is not possible to assess clearly if national welfare state contexts play a role in the effects of precarious work on mental health [64].

#### 4.1.4. Remarks on the Existing Literature

If we break down the body of literature on precarious work discussed above, the number of studies (apart from job insecurity) seems limited, both internationally and in Germany (Table 1 and Table 2). We acknowledge that in our literature review, being not systematic, we might have overlooked some relevant studies. A reason for this could be that aspects of precarious employment were not included in cohort studies outside of Scandinavia. Alternatively, such data might exist but have not been analysed due to a lack of focus on precarious work in the field of occupational health and safety [6].

### 4.2. Strengths and Limitations

A first strength of this study is that it is based on a prospective design and on a large sample of employees. Secondly, this is the first study examining a broader range of indicators of precarious work in Germany. A previous study of more than 7000 employees examined two indicators concurrently, namely job insecurity and unemployment [22]. Thirdly, we were able to control for SEP to reduce the potential confounding role of other social factors in the observed associations.

It is a limitation that the study is observational, which might introduce selection bias. Previous studies performed in countries other than Germany have shown that poor mental health prior to work entry predicts subsequent participation in the labour market [65,66,67,68]. In addition, during a work career, depressive symptoms seem to be related to the development of a poorer work environment, but to a limited degree [69,70,71,72,73]. Regarding the German context, we are not aware of the impact of such selection processes. Response bias can also play a role, especially in light of the low participation in the cohort. In the present study, cohort participation was not associated with gender, but it was with age and SEP. Among workers 30–39 years of age and unskilled workers, the response rate was lower than the response rate of older workers and that of professionals, managers and semi-professionals (Table 1). However, such differences in attrition should not alter the present findings considerably. It is also a drawback that people below the age of 31 were not invited for the study. As indicators of precarious work are more prevalent among younger workers, the present study underestimates the prevalence of precarious work [2]. We could not establish if the inclusion of younger workers would have affected the associations observed in the present study. Furthermore, the job insecurity scale is based on two items only, which resulted in a low internal consistency [74]. Some of the indictors of precarious work presented a low prevalence, limiting the statistical power of the study. This was the case for marginal part-time and fixed-term contracts among men. Moreover, we assessed SEP through the occupational level, which overlooks important aspects of SEP such as household and lifetime biographies. 

## 5. Conclusions

In line with previous studies, the present study suggests that precarious work is an important risk factor for the subsequent development of depressive symptoms among men, but not–or to a lower degree–among women. This might have implications for both practice and research.

The prevalence of precarious employment has increased in the last couple of decades, not only in Germany [75]. Such an increase might continue due to recent developments, such as the impact of the COVID–19 pandemic on the labour market. It must be noted that specific types of precarious work vary strongly across countries [49], and national contexts might play a role in their effects on health [64]. 

From the practical point of view, possible undesired health consequences of precarious employment should be taken further into account in legislation, regulation and collective agreements. Workplaces and health and safety professionals should also be aware of such undesired consequences. It is important to distinguish between forms of precarious work employment that are chosen voluntarily by workers from those that are chosen involuntary because of failed attempts to find a more stable job [61].

From the research point of view, longitudinal studies on the effects on mental health of different aspects of precarious work such as fixed-term contracts, unemployment and low wage are scarce; in addition, the impact of issues related to gender needs to be examined further [6]. Reasons for the suggested stronger effect among men should be investigated in order to understand what role economical and attitudinal aspects may play. In addition, aspects such as voluntary preferences for precarious work and health selection into precarious work should be considered. Finally, we suggest that future studies attempt to calculate fractions of unwanted health outcomes attributable to precarious work. This might further highlight the possible public health impact of this type of exposure. 

## Figures and Tables

**Figure 1 ijerph-19-03175-f001:**
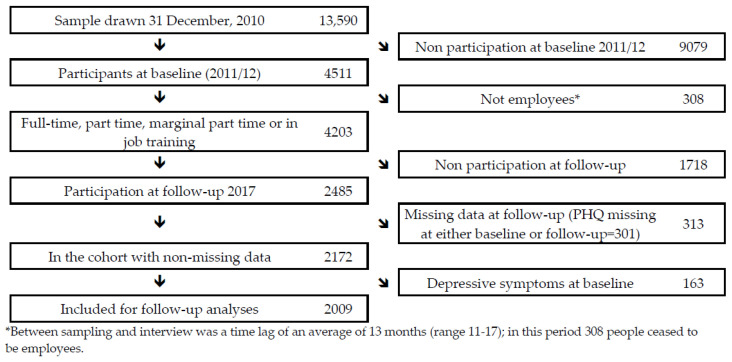
Flow diagram of participation in S-MGA’s 2012baseline and in the 2012–2017 cohort.

**Figure 2 ijerph-19-03175-f002:**
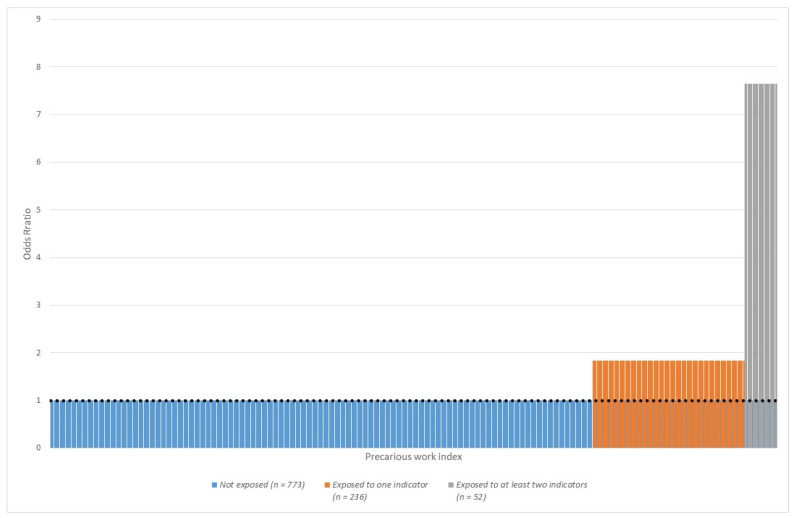
Association between precarious work index in 2012–2017 and depressive symptoms in 2017, among 1061 male employees in Germany aged 31 to 60 years, including those with depressive symptoms in 2012. Odds ratios. Precarious work index is calculated as number of exposures to (a) job insecurity at baseline, (b) low wage at baseline and (c) unemployment during follow-up. This figure illustrates the results shown in Table A9. The total area of the three bars represents all cases of depressive symptoms at follow-up. The area above the odds ratio of 1 of the two bars to the right represents those cases attributable to the elevated odds of experiencing job insecurity, unemployment or low wage. This area makes up 27% of the total area of all tree bars.

**Table 1 ijerph-19-03175-t001:** Combined indicators of precarious work as risk factors for depressive symptoms considered in six longitudinal studies.

Authors	Country	Baseline Year	*n* (Population)	Precarious Work Indicator	Outcome	Results, Overall	Results, Gender Strata
(Virtanen 2011) [19]	Sweden	1995	1005 (30 years old and over)	Temporary employment and/or job insecurity	Poor mental health (GHQ)	OR 2.33 (0.99 to 5.51)	-
(Rugulies 2010) [21]	Denmark	2000	5142 (Adult workers)	Job insecurity and/or previous prolonged unemployment	Antidepressant medication	OR 1.79 (1.15–2.79)	-
(Sirviö 2012) [23]	Finland	1997	3449 (31 year olds)	Discontinuous work history, current fixed-term and/or part-time employment	Poor mental health (HSCL–25)		M: OR 1.6 (1.1–2.3): W: OR1.4 (1.1.–1.9)
(Waenerlund 2011) [20]	Sweden	1995	985 (42 year olds)	Labour market program, on-call, seasonal, temporary agency, probationary, project employed and/or self-employed	Poor mental health	-	M: OR 2.18 (1.14–4.20); W: OR 1.79 (0.98–3.29)
(Wege 2017) [22]	Germany	2009	7354 (Adult workers)	Insecure job or long-term unemployment	Self-reported physician diagnosed depression	RR 2.30 (1.40; 3.79)	-
(Canivet 2016) [24]	Sweden	1999/2000	786 (Workers 18–34 years)	Unemployment, temporary employment, and/or perceived job insecurity	Poor mental health (GHQ–12)	RR 1.5 (1.1–2.0) Attributable fraction 18%	M: RR 2.5 (1.7–3.5) F: RR 1.8 (1.4–2.3) [42] ^1^

For more details, see Table A1. ^1^ In combination with locked job experience, i.e., being in a job without the possibility to change jobs.

**Table 2 ijerph-19-03175-t002:** Individual indicators of precarious work as risk factors for depressive symptoms considered in one longitudinal meta-analysis and seven individual longitudinal studies.

Authors	Country	*n* (Population)	Baseline Year	Precarious Work Indicator	Outcome	Results, Overall	Results, Gender Strata
Rönnblad (2019) [40]	Sweden, Norway, Denmark, Netherlands, France, Canada, US [19,21,26,27,29,30,31,35,36,37,38,39,40]	59,443 10 cohorts on adult workers and one cohort on young workers	1986–2008	Perceived job insecurity	Depression (HAD-D; SCL-CD6; MINI, CIDI-SFMD) depressive symptoms/poor mental health (CES-D; MH–5), psychological distress (GHQ–12, other), drug use (self-report, register)	Meta-analysis: OR 1.52 (1.35–1.70)	Danish Work Environment Cohort Study-poor mental health: M: 2.09 (1.04–4.20); F: 1.04 (0.62–1.74) [26]. Maastricht Cohort Study-psychological distress: M: OR 1.83 (1.33–2.51); F: 1.03 (0.62–1.71) [27]. French Santé et Itinéraire Professionnel (SIP) survey A: psychotropic drug use: job insecurity interacted with gender and was significant for men (RR = 1.38, 95% CI:1.12;1.69), but not for women [29]. B: self-reported major depressive disorder (MINI): job insecurity did not interact with gender [43].
Kim et al. (2017) [31]	South Korea	2.912 Adult workers	2012	Perceived job insecurity	Depressive symptoms (CES-D–11)	-	M: HR: 1.73 (1.16–2.59) F: HR: 1.05 (0.69–1.59)
Wege 2017 [22]	Germany	7.354 Adult workers	2009	Perceived job insecurity	Diagnosed depression	RR 1.54 (1.18; 2.01)	-
LaMontagne (2020) [28]	Australia	19,169 Adult workers	2011–2014	Job insecurity	Good mental health (MH–5)	-	M Beta: 0.34 (0.21–0.47) F: Beta: 0.09 –0.04–0.22 ^1^
Ervasti (2014) [32]	Finland	107,828	2005	Fixed-term contract Temporary employment	Sickness absence: depression	OR 1.02 (0.97–1.08)	-
Hammarström (2011) [33]	Sweden	660	1995	Fixed-term contract Temporary employment	Depressive symptoms (1-item)	OR 1.79 (1.04–3.08)	-
Quesnel-Vallée (2010) [34]	US	3.577	1994	Fixed-term contract Temporary employment	Depressive symptoms (CES-D)	ATT 1.803 (0.552–3.055)	-
Wege (2017) [22]	Germany	7.354	2009	Unemployment	Diagnosed depression	RR 1.64 (1.16; 2.31)	-
Hollander (2013) [25]	Sweden	3,284,896 (register study)	2000	Unemployment	Hospitalisation: depression	-	M: RR 2.3 (2.19–2.49); F: 1.62 (1.53–1.73)

For more details, see Table A2. ^1^ See internet appendix of the cited paper. The table is based on published analyses on baseline attrition [43] and participation in the cohort in the present study (before exclusion of participants with depressive symptoms; see Figure 1, second last box to the left).

**Table 3 ijerph-19-03175-t003:** Participation in interviews at baseline, at follow-up and in the cohort by gender, age and SES.

	Baseline (2012) Participation of the Drawn Sample ^a^; %		Follow-up (2017) Participation Among Baseline (2012) Employees ^b^, %		Cohort Participation 2017 of the Drawn Sample ^c^, %	
	*p* Value ^d^	%	*p* Value ^d^	%	*p* Value ^d^	%
GENDER	0.746		0.081		0.141	
Men		33		58		19
Women		33		60		20
AGE	0.000		0.055		0.000	
55–60		39		59		23
49–54		35		62		22
43–48		33		60		20
37–42		32		59		19
31–36		27		54		15
SEP ^e^	0.000		0.000		0.000	
Professionals, managers		38		66		25
Semi-professionals		38		62		24
Skilled workers		32		55		17
Unskilled workers		29		51		15
TOTAL		33		59		20

The table is based on published analyses on baseline attrition [43] and participation in the cohort in the present study (before exclusion of participants with depressive symptoms; see Figure 1, second last box to the left). ^a^ Fraction responding at baseline 2012 (*n* = 4511) out of the sample drawn on 31 December 2010 (*n* = 13,590). ^b^ Fraction responding at follow-up (2017) (*n* = 2485) out of the total number of baseline employees (*n* = 4203). ^c^ Fraction in the cohort (2485) out of the sample drawn on 31 December 2010 (estimated by multiplying the fraction responding at baseline by the fraction responding at follow-up). ^d^ This p value denotes to what extent each categorical variable is associated with the response (Chi2 test). ^e^ Socioeconomic position.

**Table 4 ijerph-19-03175-t004:** Description of the population of 2009 employees aged 31–60 years with non-missing information.

Variable	*n*	%	Job Insecurity 2012, %	Marginal Part–Time ^1^ 2012, %	Fixed- Term Contract 2012, %	Low Wage 2012, %	Unemployed 2012 to 2017, %
Gender 2012			0.008	<0.001	0.080	<0.001	0.007
Women	1008	50	20	10	6	16	8
Men	1001	50	20	1	4	5	5
Partner 2012			0.138	0.404	0.014	0.920	0.207
Yes	1768	87	20	5	4	11	6
No	241	13	24	4	8	11	8
Age group 2012			<0.001	0.160	0.613	0.705	0.271
31–40 years	455	22	11	4	5	10	7
41–55 years	1251	63	22	6	5	11	6
56–60 years	303	15	23	6	4	10	8
SEP 2012			<0.001	<0.001	<0.001	<0.001	0.053
Unskilled workers	109	5	34	24	11	33	8
Skilled workers	817	41	26	7	5	16	8
Semi-professionals	572	28	16	3	2	5	5
Professionals, managers	511	25	12	1	5	3	5
Job insecurity 2012				0.753	<0.001	0.003	<0.001
High	403	20		5	9	15	12
Low to medium	1606	80		5	4	10	5
Marginal part-time 2012			0.753		<0.001	<0.001	<0.001
Yes	106	5	19		12	59	15
No	1903	95	20		4	8	6
Fixed-term contract 2012			<0.001	<0.001		<0.001	<0.001
Yes	95	5	40	14		23	17
No	1914	95	19	5		10	6
Low wage 2012			0.003	<0.001	<0.001		<0.001
Yes	213	11	28	30	10		13
No	1796	89	19	2	4		5
Unemployed 2012 to 2017			<0.001	<0.001	<0.001	<0.001	
Yes	126	6	37	13	13	22	
No	1883	94	19	5	4	10	

*p* for associations using Chi2 tests are reported in the table. ^1^ Termed ‘mini- or midi-job’ in Germany.

**Table 5 ijerph-19-03175-t005:** Associations between baseline job insecurity, fixed-term contract, marginal part-time and low wage 2012 and depressive symptoms 2017 among 1001 male employees in Germany aged 31 to 60 years without depressive symptoms in 2012. Logistic regressions. Odds ratios.

	*n*	Depressive Symptoms at Follow-Up 2017 ^1^, %	Each Precarious Work Indicator Separately in the Model			Indicators of Precarious Work Mutually Adjusted		
			Adjusted for Baseline (2012) Age, Partnership Status and SEP ^2^			Adjusted for Baseline (2012) Age, Partnership Status and SEP ^2^		
			*p*	OR	95% CI	*p*	OR	95% CI
JOB INSECURITY 2012			0.003			0.015		
Low to medium	801	4		1			1	
High	200	10		2.47	1.37; 4.48		2.13	1.16; 3.92
MARGINAL PART-TIME ^3^ 2012			0.565			0.915		
No	992	5		1			1	
Yes	9	11		1.9	0.22;16.53		1.13	0.11; 11.45
FIXED-TERM 2012			0.128			0.435		
No	962	5		1			1	
Yes	39	13		2.2	0.80; 6.06		1.55	0.52; 4.66
LOW WAGE 2012			0.002			0.008		
No	952	5		1			1	
Yes	49	18		3.79	1.64; 8.72		3.22	1.36; 7.63

^1^ Total number of cases with depressive symptoms at follow-up: 54 (5%). ^2^ Socioeconomic position. ^3^ German: Minijob, Midijob.

**Table 6 ijerph-19-03175-t006:** Associations between baseline job insecurity, fixed-term contract, marginal part-time and low wage 2012 and depressive symptoms 2017 among 1008 female employees in Germany aged 31 to 60 years without depressive symptoms in 2012. Logistic regressions. Odds ratios.

	*n*	Depressive Symptoms at Follow-Up 2017 ^1^, %	Each Precarious Work Indicator Separately in the Model			Indicators of Precarious Work Mutually Adjusted		
			Adjusted for Baseline (2012) Age, Partnership Status and SEP ^2^			Adjusted for Baseline (2012) Age, Partnership Status and SEP ^2^		
			*p*	OR	95% CI	*p*	OR	95% CI
JOB INSECURITY 2012			0.142			0.209		
Low to medium	805	8		1			1	
High	203	11		1.48	0.88; 2.50		1.41	0.83; 2.40
MARGINAL PART-TIME ^3^ 2012			0.497			0.412		
No	911	9		1			1	
Yes	97	7		0.74	0.31; 1.75		0.69	0.28; 1.68
FIXED-TERM 2012			0.413			0.551		
No	952	9		1			1	
Yes	56	13		1.42	0.61; 3.32		1.30	0.55; 3.08
LOW WAGE 2012			0.595			0.478		
No	844	9		1			1	
Yes	164	9		1.19	0.63; 2.22		1.27	0.66; 2.47

^1^ Total number of cases with depressive symptoms at follow-up: 54 (5%). ^2^ Socioeconomic position. ^3^ German: Minijob, Midijob.

**Table 7 ijerph-19-03175-t007:** Associations between a 5-year experience of unemployment during follow-up (2012 to 2017) and depressive symptoms at follow-up (2017) among 1001 male and 1008 female employees in Germany aged 31 to 60 years without depressive symptoms in 2012. Logistic regressions. Odds ratios.

	Male Employees					Female Employees				
	*n*	Depressive Symptoms at Follow-Up 2017 ^1^, %	Adjusted for Baseline (2012) Age, Partnership Status and SEP ^3^			*n*	Depressive Symptoms at Follow-up 2017 ^1^, %	Adjusted for Baseline (2012) Age, Partnership Status and SEP ^3^		
			*p*	OR	95% CI			*p*	OR	95% CI
UNEMPLOYMENT 2012–2017 ^2^			0.012					0.142		
No	953	5		1		930	8		1	
Yes	48	15		3.07	1.28; 7.37	78	13		1.71	0.84; 3.49

^1^ Total number of cases with depressive symptoms at follow-up; men: 54 (5%); women: 89 (9%). ^2^ During follow-up. ^3^ Socioeconomic position.

**Table 8 ijerph-19-03175-t008:** Associations between a precarious work index ^1^ 2012–2017 and depressive symptoms 2017 among 1001 male and 1008 female employees in Germany aged 31 to 60 years without depressive symptoms in 2012. Logistic regressions. Odds ratios.

	Male Employees					Female Employees				
	*n*	Depressive Symptoms at Follow-up 2017 ^1^, %	Adjusted for Baseline Age, Partnership Status and SEP ^3^			*n*	Depressive Symptoms at Follow-up 2017 ^1^, %	Adjusted for Baseline Age, Partnership Status and SEP ^3^		
			*p*	OR	95% CI			*p*	OR	95% CI
PRECARIOUS WORK INDEX 2012–2017 ^2^			0.000					0.166		
0	748	4		1		645	8		1	
1	210	7		1.84	0.94; 3.60	288	10		1.52	0.93; 2.49
≥2	43	26		7.65	3.30; 17.73	75	11		1.74	0.76; 3.98

^1^ Total number of cases with depressive symptoms at follow-up; men: 54 (5%); women: 89 (9%). ^2^ Number of indicators of precarious work experienced: (a) job insecurity at baseline 2012, (b) low wage at baseline 2012 and (c) unemployment experience during follow-up (2012–2017). ^3^ Socioeconomic position.

## Data Availability

A scientific use file (SUF) containing both wave 1 and wave 2 of the cohort is available at the Research Data Centre of the Federal Institute of Occupational Safety and Health.

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
