# Peer review of "Precarious Work as Risk Factor for 5-Year Increase in Depressive Symptoms"

_ijerph, 2022, doi:10.3390/ijerph19063175_

Round 1
Reviewer 1 Report
Thanks for addressing my comments on the previous version. I noticed several types/grammar mistakes in this version. I recommend you pay some more attention to the use of language during the proof editing process.
My additional remarks are:
- When claiming about "limited literature" at least acknowledge that there might sources you did not identify in your searches. You included your search strategy in your paper, but since this was not strictly a systematic literature review (see respective studies what this looks like), you must be more conservative with your arguments.
- Move the Conclusions as a separate section.
Reviewer 2 Report
I appreciate your consideration of my comments and suggestions.
The article complies with what was requested.
Author Response
Please see attachment

This manuscript is a resubmission of an earlier submission. The following is a list of the peer review reports and author responses from that submission.
Round 1
Reviewer 1 Report
Comments to the paper: ijerph-1583522 Precarious work as risk factor for 5-year increase in
depressive 2 symptoms.
I appreciate the opportunity to participate in this review.
I consider it an interesting work for which I provide the following comments:
I suggest the coauthors reconsider making the study longitudinal. It is perceived to be a crosssectional analysis nested in a cohort. Although it is described that the questionnaires were applied
for 11 to 17 months, the analysis does not correspond to a longitudinal study.
I suggest contextualizing the revenues presented in Euros by projecting them more globally, for
example in dollars.
It may be worth mentioning the absence of workers under the age of 31 as an interesting factor in
the analyses. It is necessary to reconsider the analysis description considering that this is not a
longitudinal study.
I suggest extending in the discussion of results the possible sex differences consistently reported in
the scientific evidence, since this analysis essentially refers to the findings in males.
The presented calculated index is an important contribution of this research, which can be
suggested for replication in various social and labor scenarios. I suggest further discussion
of its advantages and disadvantages.
Author Response
Reviewer 1.
1.1.I appreciate the opportunity to participate in this review. I consider it an interesting work for which I provide the following comments:
AUTHORS’ ANSWER: Thank you.
1.2. I suggest the coauthors reconsider making the study longitudinal. It is perceived to be a crosssectional analysis nested in a cohort. Although it is described that the questionnaires were applied
for 11 to 17 months, the analysis does not correspond to a longitudinal study.
AUTHORS’ ANSWER: Thank you for this important comment. The study is longitudinal, employing information on precarious work from wave 1 in 2011/12 and information on depressive symptoms in 2011/12 and 2017. We now have:
- (Page 4) clarified Figure 1 by adding the appropriate years in the boxes where they were missing
- modified the abstract (Page 1), the results subsection of the text (Page 9-10) and the tables (Page 8-11 – and in the internet appendix) in order to make the longitudinal character of the analyses clearer. This was done by adding the year(s) of baseline and follow-up study waves in parentheses.
1.3. I suggest contextualizing the revenues presented in Euros by projecting them more globally, for example in dollars.
AUTHORS’ ANSWER: We now also mention the revenues in Dollars (Page 6, lines 185, 191-195). Also, we corrected the cut point value of €, as we reported the median value and not 60% of the median as we should have done. Now the right amount of € ere reported.
1.4. It may be worth mentioning the absence of workers under the age of 31 as an interesting factor in the analyses.
AUTHORS’ ANSWER: Thank you for this comment. We now highlight the age range of the population also in the title of Table 3 (description of the study sample) (Page 7). We also mention the age range explicitly in the beginning of the results and discussion sections (Page 9 line 274, Page 12 line 248). In the original manuscript, this limitation is already discussed under strengths and limitations of the study, subsection 4.2 (Page 14, line 460).
1.5. It is necessary to reconsider the analysis description considering that this is not a
longitudinal study.
AUTHORS’ ANSWER: Thank you for this comment. See our answer to comment 1.2.
1.6. I suggest extending in the discussion of results the possible sex differences consistently reported in the scientific evidence, since this analysis essentially refers to the findings in males.
AUTHORS’ ANSWER: We made some changes in the manuscript regarding this issue.
(Page 1 lines 23-24 & 27-28) In the abstract, the negative results regarding women were emphasized by adding the following sentences: ‘Among women, indicators of precarious work were not associated to depressive symptoms at follow-up’ & ‘This index was not associated with depressive symptoms among women.’
(Page 12, lines 352-3) In the first paragraph of the Discussion section, we end the first paragraph by stressing the negative results regarding women: ‘ In regard to women, however, we did not observe any significant associations between precarious work and depressive symptoms.’
(Page 12, lines 358-84) In the third paragraph at the beginning of the Discussion section, we now extend the discussion of this finding regarding men and women by also suggesting the “breadwinner” explanation. The paragraph now reads: ‘It has been suggested that uncertainty about one’s own ability to sustain a living is a key explanation for the role of precarious work in the development of depressive symptoms (de Vogli 2004; Lewchuk et al. 2008). In the present study, job insecurity and unemployment predicted depressive symptoms among men; these could be considered as the key sources of uncertainty among male employees. A number of studies carried out in European countries generally supports our finding that precarious work has a stronger effect on depressive symptoms among men than among women (Bultmann et al. 2002; Canivet et al. 2017; Hollander et al. 2013; Kim et al. 2017; LaMontagne et al. 2020; Lassalle et al. 2015; Rugulies et al. 2006; Waenerlund et al. 2011). It should be noted that in several studies the effects for women were statistically significant but of a lower size than those observed among men. There might be a number of possible explanations for such a gender difference. One could be that in Germany – as it is the case of other countries where studies about precarious work were performed – men contribute a to a higher degree than women to the household’s income (Bisello and Mascherini 2017). For this reason, precariousness can be expected to be a stronger stressor among men because of their common role as breadwinners. We are only aware of one longitudinal study that has examined such a role (Kim et al. 2017); the authors found that the risk of depressive symptoms associated with job insecurity were only elevated among men who were contributing the most to the household, but not among other men (no clear results could be obtained among women due to lack of power). Another explanation could be that men tend to identify more strongly with their work role than women, whereas women identify more with family obligations (Wiley 1991). We are not aware of more recent explanation attempts regarding work roles. As precarious work represent as a threat to the work role, it may thus affect mental health to a higher degree among men than among women. Also, it might be that men and women have different preference in relation to precarious work. For example, for some workers fixed-term contracts offer an opportunity to experience various work situations in order to gain skill and experience (Kauhanen and Nätti 2015). However, in Germany there is no gender difference regarding voluntary choice of fixed-term contracts (Eurostat 2022). We are not aware of data on voluntary choice of other types of precarious work among German employees.
(Page 12 line 388-Page 13 line 434) In the subsection “Comparison with other studies”, we now systematically discuss our and other studies’ results regarding gender differences (see also new Tables 1A and 1B on Page 3).
1.7. The presented calculated index is an important contribution of this research, which can be
suggested for replication in various social and labor scenarios. I suggest further discussion
of its advantages and disadvantages.
AUTHORS’ ANSWER: Thank you for this comment. In the part comparing our study with other measures of concurrent exposure to precarious work (section 4.1.1.), we now take this issue into account by adding the following lines: (Page 13, line 401-4) ‘Calculations of attributable fractions are advantageous as they can form a basis for prioritization of public health measures. Of course, one should take into account to what extent prevention is possible and if associations being found are causal.’ In the Conclusions subsection (4.3.), we added the following lines: (Page 14, line 492-4) ‘Finally, we suggest that future studies should attempt to calculate fractions of unwanted health outcomes attributable for precarious work. This might further highlight the possible public health importance of this issue.’
Reviewer 2 Report
The topic of this article is very important and relevant. The literature review is detailed and has a clear structure.
The authors justify the need to study the problem of precarious work as a risk factor in the formation of depressive symptoms.
The material in the article is presented logically and consistently. Research methods are justified.
The results of the study are also presented well and logically.
I would like to emphasize separately that the authors in conclusion write about some limitations and possible prospects for the development of such studies in the future.
The list of references includes 78 sources, it doesn't include an abnormal number of self-citations.
Author Response
Reviewer 2.
2.1 The topic of this article is very important and relevant.
AUTHORS’ ANSWER: Thank you.
2.2. The literature review is detailed and has a clear structure.
AUTHORS’ ANSWER: Thank you. As response to other reviewers’ comments we have added Tables 1A and 1B in the intro (Page 3) to provide an overview of previous studies. The paragraphs on previous studies in the introduction (Page 2 line 72-86) and subsection 4.1. about the comparison with earlier findings was shortened and modifiedaccordingly (Page 12 line 388-Page 13 line 426).
2.3. The authors justify the need to study the problem of precarious work as a risk factor in the formation of depressive symptoms.
AUTHORS’ ANSWER: Thank you.
2.4. The material in the article is presented logically and consistently. Research methods are justified.
AUTHORS’ ANSWER: Thank you.
2.5. The results of the study are also presented well and logically.
AUTHORS’ ANSWER: Thank you.
2.6. I would like to emphasize separately that the authors in conclusion write about some limitations and possible prospects for the development of such studies in the future.
AUTHORS’ ANSWER: Thank you.
The list of references includes 78 sources, it doesn't include an abnormal number of self-citations.
AUTHORS’ ANSWER: Thank you.
Reviewer 3 Report
The manuscript “Precarious work as risk factor for 5-year increase in depressive symptoms” studies the association between job insecurity and the development of depressive symptoms in a smaple of German employers. It is a topic of great interest, with obvious socio-political implications. The design is prospective and part of a large randomly selected sample. It analyzes different aspects of precariousness, such as job insecurity, working in a fixed-term contract, marginal part-time, low wage and unemployment. The main finding is that, only among men, job insecurity and low wage at baseline were associated with depressive symptoms at follow-up. This gender-related result is enormously suggestive and novel. Statistical analysis is adequate and well described. In summary, it seems to me a well-developed and well-written article, and one that may be of interest to readers.
I would only add some comments:
- the tables seem unconfigured, I would recommend simplifying their presentation
- the rate of non-response to the questionnaire was considerable. Of 13,590 randomly selected subjects, only 2,009 completed the follow-up. Were the sociodemographic characteristics of the responders and non-responders analyzed? I would recommend doing it.
- It is usually recommended to include the "Strengths and limitations" section at the end of the Discussion, after comparison with other studies.
- The specificity of the results in the male gender is briefly discussed. I would recommend proposing some explanatory hypothesis.
- In general, both in the abstract and in the text, the positive results are underlined. I would recommend indicating the non-significant analyzes more clearly, since they also provide valuable information.
Author Response
Reviewer 3
The manuscript “Precarious work as risk factor for 5-year increase in depressive symptoms” studies the association between job insecurity and the development of depressive symptoms in a sample of German employers. It is a topic of great interest, with obvious socio-political implications. The design is prospective and part of a large randomly selected sample. It analyses different aspects of precariousness, such as job insecurity, working in a fixed-term contract, marginal part-time, low wage and unemployment. The main finding is that, only among men, job insecurity and low wage at baseline were associated with depressive symptoms at follow-up. This gender-related result is enormously suggestive and novel. Statistical analysis is adequate and well described. In summary, it seems to me a well-developed and well-written article, and one that may be of interest to readers.
I would only add some comments:
3.1. - the tables seem unconfigured, I would recommend simplifying their presentation
AUTHORS’ ANSWER: We made two changes in the way we report the results in the tables (nos. Table 4A-6 and Appendix Tables 2A-6):
- We now only present two models, one where indicators of precarious work were included separately, and one with these indicators being mutually adjusted. In the original version, we distinguished in both cases between analysis adjusted and not adjusted for social class, but since the two yielded almost the same results, it did not add more information for the reader.
- Also, we formatted the tables so that they took less space, thus improving readability.
3.2. - the rate of non-response to the questionnaire was considerable. Of 13,590 randomly selected subjects, only 2,009 completed the follow-up. Were the sociodemographic characteristics of the responders and non-responders analyzed? I would recommend doing it.
AUTHORS’ ANSWER: Thank you for this comment. We now add a non-response analysis comparing the composition of the randomly selected subjects with the participants in the analysed cohort (new Table 2, Page 4). Accordingly we commented these results on Page 4 lines 128-31 and Page 14 lines 454-456.
3.3. - It is usually recommended to include the "Strengths and limitations" section at the end of the Discussion, after comparison with other studies.
AUTHORS’ ANSWER: (Page 13 line 442 - Page 14 line 469) We have moved this subsection at the end of the discussion section.
3.4. - The specificity of the results in the male gender is briefly discussed. I would recommend proposing some explanatory hypothesis.
(Page 12, lines 358-84) In the third paragraph at the beginning of the Discussion section, we now extend the discussion of this finding regarding men and women by also suggesting the “breadwinner” explanation. The paragraph now reads: ‘It has been suggested that uncertainty about one’s own ability to sustain a living is a key explanation for the role of precarious work in the development of depressive symptoms (de Vogli 2004; Lewchuk et al. 2008). In the present study, job insecurity and unemployment predicted depressive symptoms among men; these could be considered as the key sources of uncertainty among male employees. A number of studies carried out in European countries generally supports our finding that precarious work has a stronger effect on depressive symptoms among men than among women (Bultmann et al. 2002; Canivet et al. 2017; Hollander et al. 2013; Kim et al. 2017; LaMontagne et al. 2020; Lassalle et al. 2015; Rugulies et al. 2006; Waenerlund et al. 2011). It should be noted that in several studies the effects for women were statistically significant but of a lower size than those observed among men. There might be a number of possible explanations for such a gender difference. One could be that in Germany – as it is the case of other countries where studies about precarious work were performed – men contribute a to a higher degree than women to the household’s income (Bisello and Mascherini 2017). For this reason, precariousness can be expected to be a stronger stressor among men because of their common role as breadwinners. We are only aware of one longitudinal study that has examined such a role (Kim et al. 2017); the authors found that the risk of depressive symptoms associated with job insecurity were only elevated among men who were contributing the most to the household, but not among other men (no clear results could be obtained among women due to lack of power). Another explanation could be that men tend to identify more strongly with their work role than women, whereas women identify more with family obligations (Wiley 1991). We are not aware of more recent explanation attempts regarding work roles. As precarious work represent as a threat to the work role, it may thus affect mental health to a higher degree among men than among women. Also, it might be that men and women have different preference in relation to precarious work. For example, for some workers fixed-term contracts offer an opportunity to experience various work situations in order to gain skill and experience (Kauhanen and Nätti 2015). However, in Germany there is no gender difference regarding voluntary choice of fixed-term contracts (Eurostat 2022). We are not aware of data on voluntary choice of other types of precarious work among German employees.
3.5. - In general, both in the abstract and in the text, the positive results are underlined. I would recommend indicating the non-significant analyzes more clearly, since they also provide valuable information.
AUTHORS’ ANSWER: We made some changes in the manuscript regarding this issue.
(Page 1, line 23-24 & 27-28) In the abstract, the negative results regarding women were emphasized by adding the following sentences: ‘Among women, indicators of precarious work were not associated to depressive symptoms at follow-up’ & ‘This index was not associated with depressive symptoms among women.’
(Page 12, lines 352-3) In the first paragraph of the Discussion section, we end the first paragraph by stressing the negative results regarding women: ‘ In regard to women, however, we did not observe any significant associations between precarious work and depressive symptoms.’.
Reviewer 4 Report
Thanks for the opportunity to review your work on this important, and perhaps underestimated, topic. My comments below focus mainly on improving the clarity and communication of your work. My impression is that you refer very often to previous works (of yours) on this, which is fine. However, you need to establish a fine balance so that readers do not have to read many other papers to understand basic methodological choices or other areas of your study. Your article must be self-standing to the extent possible.
Abstract
-Focus on the results related to the objective (e.g., Line 21-22 does not seem relevant).
-The dose-response argument is not clear to the reader before reading the paper. If not a principal finding, you could remove it. Otherwise, please rephrase/elaborate.
Introduction
-Line 40. You could define "marginal part-time work" as the readers might not be familiar with the term. You explain it later in section 2.2.2, but it is better to define it here.
-Lines 39-40 and overall: You present those arrangements as enforced top-down (e.g., business to workers), but they can also be preferred by some workers for various reasons (e.g., other commitments, sense of freedom/flexibility). Please approach the topic more holistically and briefly outline possible benefits before you focus on the "undesired" side of it.
-Line 73: Please explain "..we are aware of 23 prospective studies". Was this a convenience sampling of papers or did you identify them through a (semi)systematic approach?
-Line 74: Why especially mention "non-Scandinavian origin"?
-Lines 77-78: If you did not search the literature systematically, you should avoid this assertiveness "while no prospective studies exist investigating the effects of marginal part-time"
-Lines 71-81 in general: Provide the principal findings from those studies and highlight what they did not cover, which you target with your study. You offer several details in section 4.2, but you could use that section only to refer to differences and similarities. I suggest you move the details from section 4.2 here. Then, you need to substantiate why we need your research if there is already research out there.
Methods and Materials
-Line 102: It would be more accurate to refer to 2485/4203=59% response rate. The initially targeted population had no chance to participate in the follow-up study if they had not already joined the baseline study.
-Figure 1: There are some typos in the second box from the bottom on the left part.
-Line 143: The question "What are you currently doing full-time?" seems not to match the goal. Is it a matter of translation?
-Lines 152-150: The measurement scales are unclear. Were those nominal variables?
-Lines 158-159: Unclear what you mean by "...which were recalculated as midpoints of the ranges within each response category".
-Lines 160-162; Were working hours used as continuous variables?
-Lines 164-165: The question (translation from German?) is not understood. Also, what other response options did you have?
-Lines 227-228: I am not sure whether is useful to report as it is rather expected.
-Line 229: Correlated with what?
-Tables 2A, 2B, 3 & 4 are difficult to read. You could split them into two parts each and make the results from the adjustments clearer.
Discussion
-Section 4.2.1: Minimise the text by entering a table with these data. The reader cannot follow all the figures/findings across the studies and then understand them in the light of your results.
-Section 4.2.3: Nice, but extend the discussion beyond the mere comparison of findings. For example, could be the case that female workers chose aspects you assigned as precarious work because they are loaded with family commitments, and our culture expects that males must focus more on income? You could consult with a couple of studies to support any reasonable argument.
There is no Conclusions section.
Author Response
Reviewer 4.
4.1. Thanks for the opportunity to review your work on this important, and perhaps underestimated, topic. My comments below focus mainly on improving the clarity and communication of your work.
AUTHORS’ ANSWER: Thank you.
4.2. My impression is that you refer very often to previous works (of yours) on this, which is fine. However, you need to establish a fine balance so that readers do not have to read many other papers to understand basic methodological choices or other areas of your study. Your article must be self-standing to the extent possible.
AUTHORS’ ANSWER: Thank you for this important comment. We added an attrition analysis performed for this paper specifically, with information regarding: a) baseline participation, b) follow-up participation and c) cohort participation shown in Table 2 (Page 4). Accordingly, we also changed the text in the sample subsection in the method section (Page 4 line 128-32) and the strengths and weaknesses subsection in the discussion section (Page 14 line 454-6). We now no longer refer to our own paper on workability, which contained similar information on attrition (Burr et al. 2022).
We still would like refer to (Rose et al. 2017) as this publication has more details (e.g., on sampling procedures and mode of data collection), which would have taken to much space in the present manuscript (Page 4 line 119, page 5, line 141 and 143).
4.3. Abstract
-Focus on the results related to the objective (e.g., Line 21-22 does not seem relevant).
AUTHORS’ ANSWER: We deleted the first sentence, which reads: “Women experienced precarious work more often than men” (Page 9, line 278).
4.4. -The dose-response argument is not clear to the reader before reading the paper. If not a principal finding, you could remove it. Otherwise, please rephrase/elaborate.
AUTHORS’ ANSWER: We have shortened this sentence, which now reads: (Page 1 line 25-27) “Among men, a cumulative exposure index of precarious work was associated with the development of depressive symptoms (1 indicator: 1.84; 0.94-3.60, ≥2 indicators: 7.65; 3.30-17.73).”, thus leaving out the words: “in a dose-response fashion”. Accordingly we shortened a similar sentence on Page 10, line 311).
4.5. Introduction
-Line 40. You could define "marginal part-time work" as the readers might not be familiar with the term. You explain it later in section 2.2.2, but it is better to define it here.
AUTHORS’ ANSWER: Thank you for this comment. We made two changes:
- (Page 6 line 171-179) In the variable description, we now refer to Broughton’s paper (Broughton et al. 2016), which suggested the label ‘marginal part-time work’ for the German terms Minijob and Midijob (in the variable description, we had already explained the term in the original version of the manuscript).
- We have added that the label “marginal part-time” indicates “Mini- and Midijob” in several places of the manuscript and in all the tables (Table 3-9 and Appendix Tables 2A-4B).
4.6. -Lines 39-40 and overall: You present those arrangements as enforced top-down (e.g., business to workers), but they can also be preferred by some workers for various reasons (e.g., other commitments, sense of freedom/flexibility). Please approach the topic more holistically and briefly outline possible benefits before you focus on the "undesired" side of it.
AUTHORS’ ANSWER: Thank you for this important comment. We now take this aspect up in the discussion about gender differences end of the third paragraph in the beginning of the discussion (Page 12 line 379-384). We take this up again in the conclusion subsection (Page 490-2).
4.7. -Line 73: Please explain "..we are aware of 23 prospective studies". Was this a convenience sampling of papers or did you identify them through a (semi)systematic approach?
AUTHORS’ ANSWER: Thank you for this important comment. Even if we actually used a search term ((Mental health) or (distress) or (depression) or (depressive symptoms) or (depressive disorder) or (wellbeing)) and ((Job insecurity) or (Job security) or (Precarious employment) or (marginal part time) or (minijob) or (midijob) or (mini job) or (midi job) or (fixed term employment) or (low wage)) and ((cohort) or (panel) or (longitudinal)), we have not documented the search strategy steps – therefore we abstain from using the term ‘Systematic’). We now write in the introduction (Page 2 lines 72-76):
‘Based on literature searches performed on Medline and EbscoHost in the period 2020-2021, as well as forward and backward citation searches on Web of Science and Google Scholar, we found 22 longitudinal studies and one meta-analysis covering 11 of these studies, investigating the association between precarious work and depressive symptoms (Andrea et al. 2009; Bultmann et al. 2002; Burgard et al. 2009; Canivet et al. 2016; Ervasti et al. 2014; Hammarström et al. 2011; Hollander et al. 2013; Johannessen et al. 2013; Kim et al. 2017; LaMontagne et al. 2020; Lassalle et al. 2015; Magnusson Hanson et al. 2015; Niedhammer et al. 2015; Quesnel-Vallée et al. 2010; Rönnblad et al. 2019; Rugulies et al. 2006; Rugulies et al. 2010; Sirviö et al. 2012; Virtanen et al. 2011; Waenerlund et al. 2011; Wang 2004; Wege et al. 2017) (Table 1A and 1B).’
4.8. -Line 74: Why especially mention "non-Scandinavian origin"?
AUTHORS’ ANSWER: (Page 2 line 77-79) We now use a more straightforward phrasing to avoid negations: ‘Of these, 11 were of Scandinavian origin; six were based on data from continental Europe and five on data from other continents (North America, Asia, Australia).’
4.9. -Lines 77-78: If you did not search the literature systematically, you should avoid this assertiveness "while no prospective studies exist investigating the effects of marginal part-time"
AUTHORS’ ANSWER: We still would like to keep this sentence. See answer to point 4.7.
4.10. -Lines 71-81 in general: Provide the principal findings from those studies and highlight what they did not cover, which you target with your study. You offer several details in section 4.2, but you could use that section only to refer to differences and similarities. I suggest you move the details from section 4.2 here. Then, you need to substantiate why we need your research if there is already research out there.
AUTHORS’ ANSWER: Very good point. We have added Tables 1A and 1B in the intro (Page 3) to provide an overview of previous studies. We modified accordingly the paragraphs on previous studies in the introduction (Page 2 line 72-87) and subsection 4.1. about the comparison with earlier findings (Page 12 line 385 – Page 13 line 426).
4.11. Methods and Materials
-Line 102: It would be more accurate to refer to 2485/4203=59% response rate. The initially targeted population had no chance to participate in the follow-up study if they had not already joined the baseline study.
AUTHORS’ ANSWER: Thank you for this comment. See our answer to your comment 4.2.
4.12. -Figure 1: There are some typos in the second box from the bottom on the left part.
AUTHORS’ ANSWER: We now provide a better conversion of the word file into graphics in order to avoid distortion of the box lines (Figure 1, Page 5).
4.13. -Line 143: The question "What are you currently doing full-time?" seems not to match the goal. Is it a matter of translation?
AUTHORS’ ANSWER: (Page 6, line 201-8) Yes, this translation was incorrect, thank you. We have corrected the German wording into: “What is your main activity?". The response option was also incorrect, and now it reads: “Unemployed”.
4.1. -Lines 152-150: The measurement scales are unclear. Were those nominal variables?
AUTHORS’ ANSWER: (Page 6 lines 171-179) We assume the comment regards lines 142-150. We have rephrased the paragraph so that it is clear that the variable is dichotomous.
4.15. -Lines 158-159: Unclear what you mean by "...which were recalculated as midpoints of the ranges within each response category".
AUTHORS’ ANSWER: (Page 6 lines 183-199) We have rephrased this paragraph so that all steps are transparent.
4.16. -Lines 160-162; Were working hours used as continuous variables?
AUTHORS’ ANSWER: Yes. See response to answer to previous comment 4.15.
4.17. -Lines 164-165: The question (translation from German?) is not understood. Also, what other response options did you have?
AUTHORS’ ANSWER: Thank you for this comment. We revised the translation of the question, which now reads Page 6, lines 201-208) : “What was your main activity then?"” and we corrected an error as the response option “temporary” was wrong; the right response option was “unemployed”. We also now mention that there was a range of other response options.
4.18. -Lines 227-228: I am not sure whether is useful to report as it is rather expected.
AUTHORS’ ANSWER: We have deleted the sentence (Page 9, line 280): ‘Low wage occurred seven times more often among those with marginal part-time than among those without.’
4.19. -Line 229: Correlated with what?
AUTHORS’ ANSWER: We rephrased the sentence, which now reads: “In general, all indicators of precarious work were strongly correlated with each other, ….” (Page 9, line 280).
4.20. -Tables 2A, 2B, 3 & 4 are difficult to read. You could split them into two parts each and make the results from the adjustments clearer.
AUTHORS’ ANSWER: (Table 4A-Table 6 and Appendix Tables 2A-6) We decided not to split the tables, but simplify each of them instead. We now only present two models, one where indicators of precarious work were included separately, and one with these indicators being mutually adjusted. In the original version, we distinguished in both cases between analyses adjusted and not adjusted for social class, but since the two yielded almost the same results, it did not add more information for the reader.
In addition, we formatted the tables so that they took less space, thus improving readability.
4.21. Discussion
-Section 4.2.1: Minimise the text by entering a table with these data. The reader cannot follow all the figures/findings across the studies and then understand them in the light of your results.
AUTHORS’ ANSWER: Thank you for this important comment. See answer to your comment 4.10.
4.22. -Section 4.2.3: Nice, but extend the discussion beyond the mere comparison of findings. For example, could be the case that female workers chose aspects you assigned as precarious work because they are loaded with family commitments, and our culture expects that males must focus more on income? You could consult with a couple of studies to support any reasonable argument.
AUTHORS’ ANSWER: (Page 12, lines 386-84) Thank you for this important comment. We now discuss this issue in the third paragraph in the beginning of the discussion section, where we take up three possible explanations, that is the breadwinner explanation (new – with a new reference), the work role explanation (which already was mentioned in the original manuscript), and the voluntary aspect (new, with new references).
4.23. There is no Conclusions section.
AUTHORS’ ANSWER: (Page 14, lines 470-94) We added this subsection and integrated the former ‘Perspectives’ subsection in it. In the conclusion subsection, we take up several aspects mentioned earlier in the discussion, i.e., a) those that came up when discussing gender differences in the results of this and other papers, b) the issue of attributable fractions as suggested by reviewer 1 and c) issues that emerged when discussing the existing body of research on the issue in focus.